# Minimax Time Series Prediction

**Wouter M. Koolen**
Centrum Wiskunde & Informatica
wmkoolen@cwi.nl

**Alan Malek**
UC Berkeley
malek@berkeley.edu

**Peter L. Bartlett**
UC Berkeley & QUT
bartlett@cs.berkeley.edu

**Yasin Abbasi-Yadkori**
Queensland University of Technology
yasin.abbasiyadkori@qut.edu.au

## Abstract

We consider an adversarial formulation of the problem of predicting a time series with square loss. The aim is to predict an arbitrary sequence of vectors almost as well as the best smooth comparator sequence in retrospect. Our approach allows natural measures of smoothness such as the squared norm of increments. More generally, we consider a linear time series model and penalize the comparator sequence through the energy of the implied driving noise terms. We derive the minimax strategy for all problems of this type and show that it can be implemented efficiently. The optimal predictions are linear in the previous observations. We obtain an explicit expression for the regret in terms of the parameters defining the problem. For typical, simple definitions of smoothness, the computation of the optimal predictions involves only sparse matrices. In the case of norm-constrained data, where the smoothness is defined in terms of the squared norm of the comparator's increments, we show that the regret grows as $T/\sqrt{\lambda_T}$, where $T$ is the length of the game and $\lambda_T$ is an increasing limit on comparator smoothness.

## 1 Introduction

In *time series prediction*, *tracking*, and *filtering* problems, a learner sees a stream of (possibly noisy, vector-valued) data and needs to predict the future path. One may think of robot poses, meteorological measurements, stock prices, etc. Popular stochastic models for such tasks include the auto-regressive moving average (ARMA) model in time series analysis, Brownian motion models in finance, and state space models in signal processing.

In this paper, we study the time series prediction problem in the regret framework; instead of making assumptions on the data generating process, we ask: can we predict the data sequence online almost as well as the best offline prediction method in some comparison class (in this case, offline means that the comparator only needs to model the data sequence after seeing all of it)? Our main contribution is computing the exact minimax strategy for a range of time series prediction problems. As a concrete motivating example, let us pose the simplest nontrivial such minimax problem

$$\min_{\boldsymbol{a}_1} \max_{\boldsymbol{x}_1 \in \mathcal{B}} \cdots \min_{\boldsymbol{a}_T} \max_{\boldsymbol{x}_T \in \mathcal{B}} \underbrace{\sum_{t=1}^{T} \|\boldsymbol{a}_t - \boldsymbol{x}_t\|^2}_{\text{Loss of Learner}} - \min_{\hat{\boldsymbol{a}}_1, \ldots, \hat{\boldsymbol{a}}_T} \left\{ \underbrace{\sum_{t=1}^{T} \|\hat{\boldsymbol{a}}_t - \boldsymbol{x}_t\|^2}_{\text{Loss of Comparator}} + \lambda_T \underbrace{\sum_{t=1}^{T+1} \|\hat{\boldsymbol{a}}_t - \hat{\boldsymbol{a}}_{t-1}\|^2}_{\text{Comparator Complexity}} \right\}.$$

(1)

This notion of regret is standard in online learning, going back at least to [1] in 2001, which views it as the natural generalization of $L_2$ regularization to deal with non-stationarity comparators. We offer two motivations for this regularization. First, one can interpret the complexity term as the magnitude

of the noise required to generate the comparator using a multivariate Gaussian random walk, and, generalizing slightly, as the energy of the innovations required to model the comparator using a single, fixed linear time series model (e.g. specific ARMA coefficients). Second, we can view the comparator term in Equation (1) as akin to the Lagrangian of a constrained optimization problem. Rather than competing with the comparator sequence $\hat{a}_1, \dots, \hat{a}_T$ that minimizes the cumulative loss subject to a hard constraint on the complexity term, the learner must compete with the comparator sequence that best trades off the cumulative loss and the smoothness. The Lagrange multiplier, $\lambda_T$, controls the trade-off. Notice that it is natural to allow $\lambda_T$ to grow with $T$, since that penalizes the comparator's change per round more than the loss per round.

For the particular problem (1) we obtain an efficient algorithm using amortized $O(d)$ time per round, where $d$ is the dimension of the data; there is no nasty dependence on $T$ as often happens with minimax algorithms. Our general minimax analysis extends to more advanced complexity terms. For example, we may regularize instead by higher-order smoothness (magnitude of increments of increments, etc.), or more generally, we may consider a fixed linear process and regularize the comparator by the energy of its implied driving noise terms (innovations). We also deal with arbitrary sequences of rank-one quadratic constraints on the data.

We show that the minimax algorithm is of a familiar nature; it is a linear filter, with a twist. Its coefficients are not time-invariant but instead arise from the intricate interplay between the regularization and the range of the data, combined with shrinkage. Fortunately, they may be computed in a pre-processing step by a simple recurrence. An unexpected detail of the analysis is the following. As we will show, the regret objective in (1) is a convex quadratic function of all data, and the sub-problem objectives that arise from the backward induction steps in the minimax analysis remain quadratic functions of the past. However, they may be either concave or convex. Changing direction of curvature is typically a source of technical difficulty: the minimax solution is different in either case. Quite remarkably, we show that one can determine a priori which rounds are convex and which are concave and apply the appropriate solution method in each.

We also consider what happens when the assumptions we need to make for the minimax analysis to go through are violated. We will show that the obtained minimax algorithm is in fact highly robust. Simply applying it unlicensed anyway results in adaptive regret *bounds* that scale naturally with the realized data magnitude (or, more generally, its energy).

## 1.1 Related Work

There is a rich history of tracking problems in the expert setting. In this setting, the learner has some finite number of actions to play and must select a distribution over actions to play each round in such a way as to guarantee that the loss is almost as small as the best single action in hindsight. The problem of tracking the best expert forces the learner to compare with sequences of experts (usually with some fixed number of switches). The fixed-share algorithm [2] was an early solution, but there has been more recent work [3, 4, 5, 6]. Tracking experts has been applied to other areas; see e.g. [7] for an application to sequential allocation. An extension to linear combinations of experts where the expert class is penalized by the $p$-norm of the sequence was considered in [1].

Minimax algorithms for squared Euclidean loss have been studied in several contexts such as Gaussian density estimation [8] and linear regression [9]. In [10], the authors showed that the minimax algorithm for quadratic loss is Follow the Leader (i.e. predicting the previous data mean) when the player is constrained to play in a ball around the previous data mean. Additionally, Moroshko and Krammer [11, 12] propose a weak notion of non-stationarity that allows them to apply the last-step minimax approach to a regression-like framework.

The tracking problem in the regret setting has been considered previously, e.g. [1], where the authors studied the best linear predictor with a comparison class of all sequences with bounded smoothness $\sum_t \|a_t - a_{t-1}\|^2$ and proposed a general method for converting regret bounds in the static setting to ones in the shifting setting (where the best expert is allowed to change).

**Outline** We start by presenting the formal setup in Section 2 and derive the optimal offline predictions. In Section 3 we zoom in to single-shot quadratic games, and solve these both in the convex and concave case. With this in hand, we derive the minimax solution to the time series prediction problem by backward induction in Section 4. In Section 5 we focus on the motivating problem

(1) for which we give a faster implementation and tightly sandwich the minimax regret. Section 6 concludes with discussion, conjectures and open problems.

## 2  Protocol and Offline Problem

The game protocol is described in Figure 1 and is the usual online prediction game with squared Euclidean loss. The goal of the learner is to incur small regret, that is, to predict the data almost as well as the best complexity-penalized sequence $\hat{\boldsymbol{a}}_1 \cdots \hat{\boldsymbol{a}}_T$ chosen in hindsight. Our motivating problem (1) gauged complexity by the sum of squared norms of the increments, thus encouraging smoothness. Here we generalize to complexity terms defined by a complexity matrix $\boldsymbol{K} \succeq \boldsymbol{0}$, and charge the comparator $\hat{\boldsymbol{a}}_1 \cdots \hat{\boldsymbol{a}}_T$ by $\sum_{s,t} K_{s,t} \hat{\boldsymbol{a}}_s^\mathsf{T} \hat{\boldsymbol{a}}_t$. We recover the smoothness penalty of (1) by taking $\boldsymbol{K}$ to be the $T \times T$ tridiagonal matrix

For $t = 1, 2, \ldots, T$:

- Learner predicts $\boldsymbol{a}_t \in \mathbb{R}^d$
- Environment reveals $\boldsymbol{x}_t \in \mathbb{R}^d$
- Learner suffers loss $\|\boldsymbol{a}_t - \boldsymbol{x}_t\|^2$.

Figure 1: Protocol

$$
\begin{pmatrix}
2 & -1 & & & \\
-1 & 2 & -1 & & \\
& & \ddots & & \\
& & -1 & 2 & -1 \\
& & & -1 & 2
\end{pmatrix}, \qquad (2)
$$

but we may also regularize by e.g. the sum of squared norms ($\boldsymbol{K} = \boldsymbol{I}$), the sum of norms of higher order increments, or more generally, we may consider a fixed linear process and take $\boldsymbol{K}^{1/2}$ to be the matrix that recovers the driving noise terms from the signal, and then our penalty is exactly the energy of the implied noise for that linear process. We now turn to computing the identity and quality of the best competitor sequence in hindsight.

**Theorem 1.** *For any complexity matrix $\boldsymbol{K} \succeq \boldsymbol{0}$, regularization scalar $\lambda_T \geq 0$, and $d \times T$ data matrix $\boldsymbol{X}_T = [\boldsymbol{x}_1 \cdots \boldsymbol{x}_T]$ the problem*

$$
L^* \;:=\; \min_{\hat{\boldsymbol{a}}_1, \ldots, \hat{\boldsymbol{a}}_T} \; \sum_{t=1}^{T} \|\hat{\boldsymbol{a}}_t - \boldsymbol{x}_t\|^2 + \lambda_T \sum_{s,t} K_{s,t} \hat{\boldsymbol{a}}_s^\mathsf{T} \hat{\boldsymbol{a}}_t
$$

*has linear minimizer and quadratic value given by*

$$
[\hat{\boldsymbol{a}}_1 \cdots \hat{\boldsymbol{a}}_T] \;=\; \boldsymbol{X}_T(\boldsymbol{I} + \lambda_T \boldsymbol{K})^{-1} \qquad and \qquad L^* \;=\; \mathrm{tr}\left(\boldsymbol{X}_T(\boldsymbol{I} - (\boldsymbol{I} + \lambda_T \boldsymbol{K})^{-1})\boldsymbol{X}_T^\mathsf{T}\right).
$$

*Proof.* Writing $\hat{\boldsymbol{A}} = [\hat{\boldsymbol{a}}_1 \cdots \hat{\boldsymbol{a}}_T]$ we can compactly express the offline problem as

$$
L^* \;=\; \min_{\hat{\boldsymbol{A}}} \mathrm{tr}\left((\hat{\boldsymbol{A}} - \boldsymbol{X}_T)^\mathsf{T}(\hat{\boldsymbol{A}} - \boldsymbol{X}_T) + \lambda_T \boldsymbol{K}\hat{\boldsymbol{A}}^\mathsf{T}\hat{\boldsymbol{A}}\right).
$$

The $\hat{\boldsymbol{A}}$ derivative of the objective is $2(\hat{\boldsymbol{A}} - \boldsymbol{X}_T) + 2\lambda_T \hat{\boldsymbol{A}}\boldsymbol{K}$. Setting this to zero yields the minimizer $\hat{\boldsymbol{A}} = \boldsymbol{X}_T(\boldsymbol{I} + \lambda_T \boldsymbol{K})^{-1}$. Back-substitution and simplification result in value $\mathrm{tr}\left(\boldsymbol{X}_T(\boldsymbol{I} - (\boldsymbol{I} + \lambda_T \boldsymbol{K})^{-1})\boldsymbol{X}_T^\mathsf{T}\right)$. $\qquad \square$

Note that for the choice of $\boldsymbol{K}$ in (2) computing the optimal $\hat{\boldsymbol{A}}$ can be performed in $O(dT)$ time by solving the linear system $\boldsymbol{A}(\boldsymbol{I} + \lambda_T \boldsymbol{K}_T) = \boldsymbol{X}_T$ directly. This system decomposes into $d$ (one per dimension) independent tridiagonal systems, each in $T$ (one per time step) variables, which can each be solved in linear time using Gaussian elimination.

This theorem shows that the objective of our minimax problem is a quadratic function of the data. In order to solve a $T$ round minimax problem with quadratic regret objective, we first solve simple single round quadratic games.

## 3  Minimax Single-shot Squared Loss Games

One crucial tool in the minimax analysis of our tracking problem will be solving particular single-shot min-max games. In such games, the player and adversary play prediction $\boldsymbol{a}$ and data $\boldsymbol{x}$ resulting in payoff given by the following square loss plus a quadratic in $\boldsymbol{x}$:

$$
V(\boldsymbol{a}, \boldsymbol{x}) \;:=\; \|\boldsymbol{a} - \boldsymbol{x}\|^2 + (\alpha - 1)\|\boldsymbol{x}\|^2 + 2\boldsymbol{b}^\mathsf{T}\boldsymbol{x}. \tag{3}
$$

The quadratic and linear terms in $\boldsymbol{x}$ have coefficients $\alpha \in \mathbb{R}$ and $\boldsymbol{b} \in \mathbb{R}^d$. Note that $V(\boldsymbol{a}, \boldsymbol{x})$ is convex in $\boldsymbol{a}$ and either convex or concave in $\boldsymbol{x}$ as decided by the sign of $\alpha$. The following result, proved in Appendix B.1 and illustrated for $\|\boldsymbol{b}\| = 1$ by the figure to the right, gives the minimax analysis for both cases.

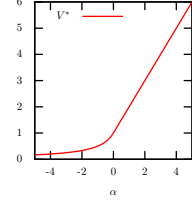

**Theorem 2.** *Let $V(\boldsymbol{a}, \boldsymbol{x})$ be as in (3). If $\|\boldsymbol{b}\| \leq 1$, then the minimax problem*

$$V^* := \min_{\boldsymbol{a} \in \mathbb{R}^d} \max_{\boldsymbol{x} \in \mathbb{R}^d : \|\boldsymbol{x}\| \leq 1} V(\boldsymbol{a}, \boldsymbol{x})$$

*has value* $V^* = \begin{cases} \dfrac{\|\boldsymbol{b}\|^2}{1-\alpha} & \text{if } \alpha \leq 0, \\ \|\boldsymbol{b}\|^2 + \alpha & \text{if } \alpha \geq 0, \end{cases}$ *and minimizer* $\boldsymbol{a} = \begin{cases} \dfrac{\boldsymbol{b}}{1-\alpha} & \text{if } \alpha \leq 0, \\ \boldsymbol{b} & \text{if } \alpha \geq 0. \end{cases}$ (4)

We also want to look at the performance of this strategy when we do not impose the norm bound $\|\boldsymbol{x}\| \leq 1$ nor make the assumption $\|\boldsymbol{b}\| \leq 1$. By evaluating (3) we obtain an adaptive expression that scales with the actual norm $\|\boldsymbol{x}\|^2$ of the data.

**Theorem 3.** *Let $\boldsymbol{a}$ be the strategy from (4). Then, for any data $\boldsymbol{x} \in \mathbb{R}^d$ and any $\boldsymbol{b} \in \mathbb{R}^d$,*

$$V(\boldsymbol{a}, \boldsymbol{x}) = \frac{\|\boldsymbol{b}\|^2}{1-\alpha} + \alpha \left\| \frac{\boldsymbol{b}}{1-\alpha} - \boldsymbol{x} \right\|^2 \leq \frac{\|\boldsymbol{b}\|^2}{1-\alpha} \qquad \text{if } \alpha \leq 0, \text{ and}$$

$$V(\boldsymbol{a}, \boldsymbol{x}) = \|\boldsymbol{b}\|^2 + \alpha \|\boldsymbol{x}\|^2 \qquad \text{if } \alpha \geq 0.$$

These two theorems point out that the strategy in (4) is amazingly versatile. The former theorem establishes minimax optimality under data constraint $\|\boldsymbol{x}\| \leq 1$ assuming that $\|\boldsymbol{b}\| \leq 1$. Yet the latter theorem tells us that, even without constraints and assumptions, this strategy is still an extremely useful heuristic. For its actual regret is bounded by the minimax regret we would have incurred if we would have known the scale of the data $\|\boldsymbol{x}\|$ (and $\|\boldsymbol{b}\|$) in advance. The norm bound we imposed in the derivation induces the complexity measure for the data to which the strategy adapts. This robustness property will extend to the minimax strategy for time series prediction.

Finally, it remains to note that we present the theorems in the canonical case. Problems with a constraint of the form $\|\boldsymbol{x} - \boldsymbol{c}\| \leq \beta$ may be canonized by re-parameterizing by $\boldsymbol{x}' = \frac{\boldsymbol{x}-\boldsymbol{c}}{\beta}$ and $\boldsymbol{a}' = \frac{\boldsymbol{a}-\boldsymbol{c}}{\beta}$ and scaling the objective by $\beta^{-2}$. We find

**Corollary 4.** *Fix $\beta \geq 0$ and $\boldsymbol{c} \in \mathbb{R}^d$. Let $V^*(\alpha, \boldsymbol{b})$ denote the minimax value from (4) with parameters $\alpha, \boldsymbol{b}$. If $\|(\alpha - 1)\boldsymbol{c} + \boldsymbol{b}\| \leq \beta$ then*

$$\min_{\boldsymbol{a}} \max_{\boldsymbol{x} : \|\boldsymbol{x} - \boldsymbol{c}\| \leq \beta} V(\boldsymbol{a}, \boldsymbol{x}) = \beta^2 V^* \left( \alpha, \frac{(\alpha - 1)\boldsymbol{c} + \boldsymbol{b}}{\beta} \right) + 2\boldsymbol{b}^\mathsf{T} \boldsymbol{c} + (\alpha - 1)\|\boldsymbol{c}\|^2.$$

With this machinery in place, we continue the minimax analysis of time series prediction problems.

## 4 Minimax Time Series Prediction

In this section, we give the minimax solution to the online prediction problem. Recall that the evaluation criterion, the regret, is defined by

$$\mathcal{R} := \sum_{t=1}^{T} \|\boldsymbol{a}_t - \boldsymbol{x}_t\|^2 - \min_{\hat{\boldsymbol{a}}_1, \dots, \hat{\boldsymbol{a}}_T} \sum_{t=1}^{T} \|\hat{\boldsymbol{a}}_t - \boldsymbol{x}_t\|^2 + \lambda_T \operatorname{tr}\left(\boldsymbol{K} \hat{\boldsymbol{A}}^\mathsf{T} \hat{\boldsymbol{A}}\right)$$ (5)

where $\boldsymbol{K} \succeq \boldsymbol{0}$ is a fixed $T \times T$ matrix measuring the complexity of the comparator sequence. Since all the derivations ahead will be for a fixed $T$, we drop the $T$ subscript on the $\lambda$. We study the minimax problem

$$\mathcal{R}^* := \min_{\boldsymbol{a}_1} \max_{\boldsymbol{x}_1} \cdots \min_{\boldsymbol{a}_T} \max_{\boldsymbol{x}_T} \mathcal{R}$$ (6)

under the constraint on the data that $\|\boldsymbol{X}_t \boldsymbol{v}_t\| \leq 1$ in each round $t$ for some fixed sequence $\boldsymbol{v}_1, \dots \boldsymbol{v}_T$ such that $\boldsymbol{v}_t \in \mathbb{R}^t$. This constraint generalizes the norm bound constraint from the motivating problem (1), which is recovered by taking $\boldsymbol{v}_t = \boldsymbol{e}_t$. This natural generalization allows us to also consider bounded norms of increments, bounded higher order discrete derivative norms etc.

We compute the minimax regret and get an expression for the minimax algorithm. We show that, at any point in the game, the value is a quadratic function of the past samples and the minimax algorithm is linear: it always predicts with a weighted sum of all past samples.

Most intriguingly, the value function can either be a convex or concave quadratic in the last data point, depending on the regularization. We saw in the previous section that these two cases require a different minimax solution. It is therefore an extremely fortunate fact that the particular case we find ourselves in at each round is *not* a function of the past data, but just a property of the problem parameters $\boldsymbol{K}$ and $\boldsymbol{v}_t$.

We are going to solve the sequential minimax problem (6) one round at a time. To do so, it is convenient to define the value-to-go of the game from any state $\boldsymbol{X}_t = [\boldsymbol{x}_1 \cdots \boldsymbol{x}_t]$ recursively by

$$V(\boldsymbol{X}_T) := -L^* \qquad \text{and} \qquad V(\boldsymbol{X}_{t-1}) := \min_{\boldsymbol{a}_t} \max_{\boldsymbol{x}_t : \|\boldsymbol{X}_t \boldsymbol{v}_t\| \le 1} \|\boldsymbol{a}_t - \boldsymbol{x}_t\|^2 + V(\boldsymbol{X}_t).$$

We are interested in the minimax algorithm and minimax regret $\mathcal{R}^* = V(\boldsymbol{X}_0)$. We will show that the minimax value and strategy are a quadratic and linear function of the observations. To express the value and strategy and state the necessary condition on the problem, we will need a series of scalars $d_t$ and matrices $\boldsymbol{R}_t \in \mathbb{R}^{t \times t}$ for $t = 1, \ldots, T$, which, as we will explain below, arises naturally from the minimax analysis. The matrices, which depend on the regularization parameter $\lambda$, comparator complexity matrix $\boldsymbol{K}$ and data constraints $\boldsymbol{v}_t$, are defined recursively back-to-front. The base case is $\boldsymbol{R}_T := (\boldsymbol{I} + \lambda_T \boldsymbol{K})^{-1}$. Using the convenient abbreviations $\boldsymbol{v}_t = w_t \begin{pmatrix} \boldsymbol{u}_t \\ 1 \end{pmatrix}$ and $\boldsymbol{R}_t = \begin{pmatrix} \boldsymbol{A}_t & \boldsymbol{b}_t \\ \boldsymbol{b}_t^\mathsf{T} & c_t \end{pmatrix}$ we then recursively define $\boldsymbol{R}_{t-1}$ and set $d_t$ by

$$\boldsymbol{R}_{t-1} := \boldsymbol{A}_t + (\boldsymbol{b}_t - c_t \boldsymbol{u}_t)(\boldsymbol{b}_t - c_t \boldsymbol{u}_t)^\mathsf{T} - c_t \boldsymbol{u}_t \boldsymbol{u}_t^\mathsf{T}, \qquad d_t := \frac{c_t}{w_t^2} \qquad \text{if } c_t \ge 0, \qquad (7a)$$

$$\boldsymbol{R}_{t-1} := \boldsymbol{A}_t + \frac{\boldsymbol{b}_t \boldsymbol{b}_t^\mathsf{T}}{1 - c_t}, \qquad\qquad\qquad d_t := 0 \qquad\quad \text{if } c_t \le 0. \qquad (7b)$$

Using this recursion for $d_t$ and $\boldsymbol{R}_t$, we can perform the exact minimax analysis under a certain condition on the interplay between the data constraint and the regularization. We then show below that the obtained algorithm has a condition-free data-dependent regret bound.

**Theorem 5.** *Assume that $\boldsymbol{K}$ and $\boldsymbol{v}_t$ are such that any data sequence $\boldsymbol{X}_T$ satisfying the constraint $\|\boldsymbol{X}_t \boldsymbol{v}_t\| \le 1$ for all rounds $t \le T$ also satisfies $\big\|\boldsymbol{X}_{t-1}\big((c_t - 1)\boldsymbol{u}_t - \boldsymbol{b}_t\big)\big\| \le 1/w_t$ for all rounds $t \le T$. Then the minimax value of and strategy for problem (6) are given by*

$$V(\boldsymbol{X}_t) = \operatorname{tr}\left(\boldsymbol{X}_t (\boldsymbol{R}_t - \boldsymbol{I}) \boldsymbol{X}_t^\mathsf{T}\right) + \sum_{s=t+1}^{T} d_s \qquad \text{and} \qquad \boldsymbol{a}_t = \boldsymbol{X}_{t-1} \begin{cases} \frac{\boldsymbol{b}_t}{1 - c_t} & \text{if } c_t \le 0, \\ \boldsymbol{b}_t - c_t \boldsymbol{u}_t & \text{if } c_t \ge 0, \end{cases}$$

In particular, this shows that the minimax regret (6) is given by $\mathcal{R}^* = \sum_{t=1}^{T} d_t$.

*Proof.* By induction. The base case $V(\boldsymbol{X}_T)$ is Theorem 1. For any $t < T$ we apply the definition of $V(\boldsymbol{X}_{t-1})$ and the induction hypothesis to get

$$V(\boldsymbol{X}_{t-1}) = \min_{\boldsymbol{a}_t} \max_{\boldsymbol{x}_t : \|\boldsymbol{X}_t \boldsymbol{v}_t\| \le 1} \|\boldsymbol{a}_t - \boldsymbol{x}_t\|^2 + \operatorname{tr}\left(\boldsymbol{X}_t (\boldsymbol{R}_t - \boldsymbol{I}) \boldsymbol{X}_t^\mathsf{T}\right) + \sum_{s=t+1}^{T} d_s$$

$$= \operatorname{tr}(\boldsymbol{X}_{t-1}(\boldsymbol{A}_t - \boldsymbol{I}) \boldsymbol{X}_{t-1}^\mathsf{T}) + \sum_{s=t+1}^{T} d_t + C$$

where we abbreviated

$$C := \min_{\boldsymbol{a}_t} \max_{\boldsymbol{x}_t : \|\boldsymbol{X}_t \boldsymbol{v}_t\| \le 1} \|\boldsymbol{a}_t - \boldsymbol{x}_t\|^2 + (c_t - 1)\boldsymbol{x}_t^\mathsf{T} \boldsymbol{x}_t + 2\boldsymbol{x}_t^\mathsf{T} \boldsymbol{X}_{t-1} \boldsymbol{b}_t.$$

Without loss of generality, assume $w_t > 0$. Now, as $\|\boldsymbol{X}_t \boldsymbol{v}_t\| \le 1$ iff $\|\boldsymbol{X}_{t-1} \boldsymbol{u}_t + \boldsymbol{x}_t\| \le 1/w_t$, application of Corollary 4 with $\alpha = c_t$, $\boldsymbol{b} = \boldsymbol{X}_{t-1} \boldsymbol{b}_t$, $\beta = 1/w_t$ and $\boldsymbol{c} = -\boldsymbol{X}_{t-1} \boldsymbol{u}_t$ followed by Theorem 2 results in optimal strategy

$$\boldsymbol{a}_t = \begin{cases} \frac{\boldsymbol{X}_{t-1} \boldsymbol{b}_t}{1 - c_t} & \text{if } c_t \le 0, \\ -c_t \boldsymbol{X}_{t-1} \boldsymbol{u}_t + \boldsymbol{X}_{t-1} \boldsymbol{b}_t & \text{if } c_t \ge 0. \end{cases}$$

and value

$$C = (c_t-1)\|\boldsymbol{X}_{t-1}\boldsymbol{u}_t\|^2 - 2\boldsymbol{b}_t^\mathsf{T}\boldsymbol{X}_{t-1}^\mathsf{T}\boldsymbol{X}_{t-1}\boldsymbol{u}_t + \begin{cases} \left\|\boldsymbol{X}_{t-1}\big((c_t-1)\boldsymbol{u}_t - \boldsymbol{b}_t\big)\right\|^2 / (1-c_t) & \text{if } c_t \le 0, \\ \left\|\boldsymbol{X}_{t-1}\big((c_t-1)\boldsymbol{u}_t - \boldsymbol{b}_t\big)\right\|^2 + c_t/w_t^2 & \text{if } c_t \ge 0, \end{cases}$$

Expanding all squares and rearranging (cycling under the trace) completes the proof. $\qquad\square$

On the one hand, from a technical perspective the condition of Theorem 5 is rather natural. It guarantees that the prediction of the algorithm will fall within the constraint imposed on the data. (If it would not, we could benefit by clipping the prediction. This would be guaranteed to reduce the loss, and it would wreck the backwards induction.) Similar clipping conditions arise in the minimax analyses for linear regression [9] and square loss prediction with Mahalanobis losses [13].

In practice we typically do not have a hard bound on the data. Sill, by running the above minimax algorithm obtained for data complexity *bounds* $\|\boldsymbol{X}_t\boldsymbol{v}_t\| \le 1$, we get an adaptive regret bound that scales with the *actual data complexity* $\|\boldsymbol{X}_t\boldsymbol{v}_t\|^2$, as can be derived by replacing the application of Theorem 2 in the proof of Theorem 5 by an invocation of Theorem 3.

**Theorem 6.** *Let $\boldsymbol{K} \succeq \boldsymbol{0}$ and $\boldsymbol{v}_t$ be arbitrary. The minimax algorithm obtained in Theorem 5 keeps the regret (5) bounded by $\mathcal{R} \le \sum_{t=1}^{T} d_t \|\boldsymbol{X}_t\boldsymbol{v}_t\|^2$ for any data sequence $\boldsymbol{X}_T$.*

### 4.1 Computation, sparsity

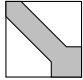

In the important special case (typical application) where the regularization $\boldsymbol{K}$ and data constraint $\boldsymbol{v}_t$ are encoding some order of smoothness, we find that $\boldsymbol{K}$ is banded diagonal and $\boldsymbol{v}_t$ only has a few tail non-zero entries. It hence is the case that $\boldsymbol{R}_{T-1}^{-1} = \boldsymbol{I} + \lambda\boldsymbol{K}$ is sparse. We now argue that the recursive updates (7) preserve sparsity of the inverse $\boldsymbol{R}_t^{-1}$. In Appendix C we derive an update for $\boldsymbol{R}_{t-1}^{-1}$ in terms of $\boldsymbol{R}_t^{-1}$. For computation it hence makes sense to tabulate $\boldsymbol{R}_t^{-1}$ directly. We now argue (proof in Appendix B.2) that all $\boldsymbol{R}_t^{-1}$ are sparse.

**Theorem 7.** *Say the $\boldsymbol{v}_t$ are $V$-sparse (all but their tail $V$ entries are zero). And say that $\boldsymbol{K}$ is $D$-banded (all but the the main and $D-1$ adjacent diagonals to either side are zero). Then each $\boldsymbol{R}_t^{-1}$ is the sum of the $D$-banded matrix $\boldsymbol{I} + \lambda\boldsymbol{K}_{1:t,1:t}$ and a $(D+V-2)$-blocked matrix (i.e. all but the lower-right block of size $D+V-2$ is zero).*

So what does this sparsity argument buy us? We only need to maintain the original $D$-banded matrix $\boldsymbol{K}$ and the $(D+V-2)^2$ entries of the block perturbation. These entries can be updated backwards from $t = T, \ldots, 1$ in $\mathcal{O}((D+V-2)^3)$ time per round using block matrix inverses. This means that the run-time of the entire pre-processing step is *linear* in $T$. For updates and prediction we need $c_t$ and $\boldsymbol{b}_t$, which we can compute using Gaussian elimination from $\boldsymbol{R}_t^{-1}$ in $O(t(D+V))$ time. In the next section we will see a special case in which we can update and predict in constant time.

## 5 Norm-bounded Data with Increment Squared Regularization

We return to our motivating problem (1) with complexity matrix $\boldsymbol{K} = \boldsymbol{K}_T$ given by (2) and norm constrained data, i.e. $\boldsymbol{v}_t = \boldsymbol{e}_t$. We show that the $\boldsymbol{R}_t$ matrices are very simple: their inverse is $\boldsymbol{I} + \lambda\boldsymbol{K}_t$ with its lower-right entry perturbed. Using this, we show that the prediction is a linear combination of the past observations with weights decaying exponentially backward in time. We derive a constant-time update equation for the minimax prediction and tightly sandwich the regret.

Here, we will calculate a few quantities that will be useful throughout this section. The inverse $(\boldsymbol{I} + \lambda\boldsymbol{K}_T)^{-1}$ can be computed in closed form as a direct application of the results in [14]:

**Lemma 8.** *Recall that $\sinh(x) = \frac{e^x - e^{-x}}{2}$ and $\cosh(x) = \frac{e^x + e^{-x}}{2}$. For any $\lambda \ge 0$:*

$$(\boldsymbol{I} + \lambda\boldsymbol{K}_T)_{i,j}^{-1} = \frac{\cosh\big((T+1-|i-j|)\nu\big) - \cosh\big((T+1-i-j)\nu\big)}{2\lambda\sinh(\nu)\sinh\big((T+1)\nu\big)},$$

*where $\nu = \cosh^{-1}\left(1 + \frac{1}{2\lambda}\right)$.*

We need some control on this inverse. We will use the abbreviations

$$z_t := (I + \lambda K_t)^{-1} e_t, \tag{8}$$

$$h_t := e_t^\mathsf{T} (I + \lambda K_t)^{-1} e_t = e_t^\mathsf{T} z_t, \text{ and} \tag{9}$$

$$h := \frac{2}{1 + 2\lambda + \sqrt{1 + 4\lambda}}. \tag{10}$$

We now show that these quantities are easily computable (see Appendix B for proofs).

**Lemma 9.** *Let $\nu$ be as in Lemma 8. Then, we can write*

$$h_t = \frac{1 - (\lambda h)^{2t}}{1 - (\lambda h)^{2t+2}} h,$$

*and $\lim_{t \to \infty} h_t = h$ from below, exponentially fast.*

A direct application of block matrix inversion (Lemma 12) results in

**Lemma 10.** *We have*

$$h_t = \frac{1}{1 + 2\lambda - \lambda^2 h_{t-1}} \qquad and \qquad z_t = h_t \begin{pmatrix} \lambda z_{t-1} \\ 1 \end{pmatrix}.$$

Intriguingly, following the optimal algorithm for all $T$ rounds can be done in $O(Td)$ computation and $O(d)$ memory. These resource requirements are surprising as playing weighted averages typically requires $O(T^2 d)$. We found that the weighted averages are similar between rounds and can be updated cheaply.

We are now ready to state the main result of this section, proved in Appendix B.3.

**Theorem 11.** *Let $z_t$ and $h_t$ be as in* (8) *and $K_t$ as in* (2). *For the minimax problem* (1) *we have*

$$R_t^{-1} = I + \lambda K_t + \gamma_t e_t e_t^\mathsf{T}$$

*and the minimax prediction in round $t$ is given by*

$$a_t = \lambda c_t X_{t-1} z_{t-1}$$

*where $\gamma_t = \frac{1}{c_t} - \frac{1}{h_t}$ and $c_t$ satisfy the recurrence $c_T = h_T$ and $c_{t-1} = h_{t-1} + \lambda^2 h_{t-1}^2 c_t (1 + c_t)$.*

## 5.1 Implementation

Theorem 11 states that the minimax prediction is $a_t = \lambda c_t X_{t-1} z_{t-1}$. Using Lemma 10, we can derive an incremental update for $a_t$ by defining $a_1 = 0$ and

$$a_{t+1} = \lambda c_{t+1} X_t z_t = \lambda c_{t+1} [X_{t-1} \ x_t] h_t \begin{pmatrix} \lambda z_{t-1} \\ 1 \end{pmatrix} = \lambda c_{t+1} h_t (X_{t-1} \lambda z_{t-1} + x_t)$$

$$= \lambda c_{t+1} h_t \left( \frac{a_t}{c_t} + x_t \right).$$

This means we can predict in constant time $O(d)$ per round.

## 5.2 Lower Bound

By Theorem 5, using that $w_t = 1$ so that $d_t = c_t$, the minimax regret equals $\sum_{t=1}^T c_t$. For convenience, we define $r_t := 1 - (\lambda_T h)^{2t}$ (and $r_{T+1} = 1$) so that $h_t = h r_t / r_{t+1}$. We can obtain a lower bound on $c_t$ from the expression given in Theorem 11 by ignoring the (positive) $c_t^2$ term to obtain: $c_{t-1} \geq h_{t-1} + \lambda_T^2 h_{t-1}^2 c_t$. By unpacking this lower bound recursively, we arrive at

$$c_t \geq h \sum_{k=t}^T (\lambda_T h)^{2(k-t)} \frac{r_t^2}{r_k r_{k+1}}.$$

Since $r_t^2/(r_i r_{i+1})$ is a decreasing function in $i$ for every $t$, we have $\frac{r_t^2}{r_i r_{i+1}} \geq \frac{r_t}{r_{t+1}}$ which leads to

$$\sum_{t=1}^{T} c_t \geq h \sum_{t=1}^{T} \sum_{k=t}^{T} (\lambda_T h)^{2(k-t)} \frac{r_t}{r_{t+1}} \geq h \int_0^{T-1} \int_{t+1}^{T} (\lambda_T h)^{2(k-t)} \frac{r_t}{r_{t+1}} dk\, dt = \Omega\left(-\frac{hT}{2\log(\lambda_T h)}\right)$$

where we have exploited the fact that the integrand is monotonic and concave in $k$ and monotonic and convex in $t$ to lower bound the sums with an integral. See Claim 14 in the appendix for more details. Since $-\log(\lambda_T h) = O(1/\sqrt{\lambda_T})$ and $h = \Omega(1/\lambda_T)$, we have that $\sum_{t=1}^{T} c_t = \Omega(\frac{T}{\sqrt{\lambda_T}})$, matching the upper bound below.

### 5.3 Upper Bound

As $h \geq h_t$, the alternative recursion $c'_{T+1} = 0$ and $c'_{t-1} = h + \lambda^2 h^2 c'_t(1 + c'_t)$ satisfies $c'_t \geq c_t$. A simple induction [1] shows that $c'_t$ is increasing with decreasing $t$, and it must hence have a limit. This limit is a fixed-point of $c \mapsto h + \lambda^2 h^2 c(1 + c)$. This results in a quadratic equation, which has two solutions. Our starting point $c'_{T+1} = 0$ lies below the half-way point $\frac{1 - \lambda^2 h^2}{2\lambda^2 h^2} > 0$, so the sought limit is the smaller solution:

$$c = \frac{-\lambda^2 h^2 + 1 - \sqrt{(\lambda^2 h^2 - 1)^2 - 4\lambda^2 h^3}}{2\lambda^2 h^2}.$$

This is monotonic in $h$. Plugging in the definition of $h$, we find

$$c = \frac{\sqrt{4\lambda + 1}(2\lambda + 1) + 4\lambda + 1 - \sqrt{2}\sqrt{2\lambda\left(\lambda\left(2\sqrt{4\lambda + 1} + 7\right) + 3\sqrt{4\lambda + 1} + 4\right) + \sqrt{4\lambda + 1} + 1}}{4\lambda^2}.$$

Series expansion around $\lambda \to \infty$ results in $c \leq (1 + \lambda)^{-1/2}$. So all in all, the bound is

$$\mathcal{R}^* = \mathcal{O}\left(\frac{T}{\sqrt{1 + \lambda_T}}\right),$$

where we have written the explicit $T$ dependence of $\lambda$. As discussed in the introduction, allowing $\lambda_T$ to grow with $T$ is natural and necessary for sub-linear regret. If $\lambda_T$ were constant, the regret term and complexity term would grow with $T$ at the same rate, effectively forcing the learner to compete with sequences that could track the $x_t$ sequence arbitrarily well.

## 6 Discussion

We looked at obtaining the minimax solution to simple tracking/filtering/time series prediction problems with square loss, square norm regularization and square norm data constraints. We obtained a computational method to get the minimax result. Surprisingly, the problem turns out to be a mixture of per-step quadratic minimax problems that can be either concave or convex. These two problems have different solutions. Since the type of problem that is faced in each round is *not* a function of the past data, but only of the regularization, the coefficients of the value-to-go function can still be computed recursively. However, extending the analysis beyond quadratic loss and constraints is difficult; the self-dual property of the 2-norm is central to the calculations.

Several open problems arise. The stability of the coefficient recursion is so far elusive. For the case of norm bounded data, we found that the $c_t$ are positive and essentially constant. However, for higher order smoothness constraints on the data (norm bounded increments, increments of increments, ...) the situation is more intricate. We find negative $c_t$ and oscillating $c_t$, both diminishing and increasing. Understanding the behavior of the minimax regret and algorithm as a function of the regularization $K$ (so that we can tune $\lambda$ appropriately) is an intriguing and elusive open problem.

#### Acknowledgments

We gratefully acknowledge the support of the NSF through grant CCF-1115788, and of the Australian Research Council through an Australian Laureate Fellowship (FL110100281) and through the ARC Centre of Excellence for Mathematical and Statistical Frontiers. Thanks also to the Simons Institute for the Theory of Computing Spring 2015 Information Theory Program.

## Footnotes

[1] For the base case, $c_{T+1} = 0 \leq c_T = h$. Then $c'_{t-1} = h + \lambda^2 h^2 c'_t(1 + c'_t) \geq h + \lambda^2 h^2 c'_{t+1}(1 + c'_{t+1}) = c'_t$.

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
