[Supplementary Material]

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

$$\boldsymbol{z}_t := (\boldsymbol{I} + \lambda \boldsymbol{K}_t)^{-1} \boldsymbol{e}_t, \tag{8}$$

$$h_t := \boldsymbol{e}_t^\mathsf{T} (\boldsymbol{I} + \lambda \boldsymbol{K}_t)^{-1} \boldsymbol{e}_t = \boldsymbol{e}_t^\mathsf{T} \boldsymbol{z}_t, \text{ and} \tag{9}$$

$$h := \frac{2}{1 + 2\lambda + \sqrt{1 + 4\lambda}}. \tag{10}$$

We now show that these quantities are easily computable (see Appendix B for proofs).

**Lemma 9.** *Let $\nu$ be as in Lemma 8. Then, we can write*

$$h_t = \frac{1 - (\lambda h)^{2t}}{1 - (\lambda h)^{2t+2}} h,$$

*and $\lim_{t \to \infty} h_t = h$ from below, exponentially fast.*

A direct application of block matrix inversion (Lemma 12) results in

**Lemma 10.** *We have*

$$h_t = \frac{1}{1 + 2\lambda - \lambda^2 h_{t-1}} \qquad and \qquad \boldsymbol{z}_t = h_t \begin{pmatrix} \lambda \boldsymbol{z}_{t-1} \\ 1 \end{pmatrix}.$$

Intriguingly, following the optimal algorithm for all $T$ rounds can be done in $O(Td)$ computation and $O(d)$ memory. These resource requirements are surprising as playing weighted averages typically requires $O(T^2 d)$. We found that the weighted averages are similar between rounds and can be updated cheaply.

We are now ready to state the main result of this section, proved in Appendix B.3.

**Theorem 11.** *Let $\boldsymbol{z}_t$ and $h_t$ be as in (8) and $\boldsymbol{K}_t$ as in (2). For the minimax problem (1) we have*

$$\boldsymbol{R}_t^{-1} = \boldsymbol{I} + \lambda \boldsymbol{K}_t + \gamma_t \boldsymbol{e}_t \boldsymbol{e}_t^\mathsf{T}$$

*and the minimax prediction in round $t$ is given by*

$$\boldsymbol{a}_t = \lambda c_t \boldsymbol{X}_{t-1} \boldsymbol{z}_{t-1}$$

*where $\gamma_t = \frac{1}{c_t} - \frac{1}{h_t}$ and $c_t$ satisfy the recurrence $c_T = h_T$ and $c_{t-1} = h_{t-1} + \lambda^2 h_{t-1}^2 c_t (1 + c_t)$.*

### 5.1 Implementation

Theorem 11 states that the minimax prediction is $\boldsymbol{a}_t = \lambda c_t \boldsymbol{X}_{t-1} \boldsymbol{z}_{t-1}$. Using Lemma 10, we can derive an incremental update for $\boldsymbol{a}_t$ by defining $\boldsymbol{a}_1 = \boldsymbol{0}$ and

$$\boldsymbol{a}_{t+1} = \lambda c_{t+1} \boldsymbol{X}_t \boldsymbol{z}_t = \lambda c_{t+1} [\boldsymbol{X}_{t-1} \ \boldsymbol{x}_t] h_t \begin{pmatrix} \lambda \boldsymbol{

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

## A  Technical lemmas

Here, we gather the version of block matrix inversion and Sherman Morrison formulas used previously.

**Lemma 12** (Block Matrix Inverse). *We have*

$$\begin{pmatrix} \boldsymbol{A} & \boldsymbol{b} \\ \boldsymbol{b}^\mathsf{T} & c \end{pmatrix}^{-1} = \begin{pmatrix} \boldsymbol{A}^{-1} + \boldsymbol{A}^{-1}\boldsymbol{b}(c - \boldsymbol{b}^\mathsf{T}\boldsymbol{A}^{-1}\boldsymbol{b})^{-1}\boldsymbol{b}^\mathsf{T}\boldsymbol{A}^{-1} & -\boldsymbol{A}^{-1}\boldsymbol{b}(c - \boldsymbol{b}^\mathsf{T}\boldsymbol{A}^{-1}\boldsymbol{b})^{-1} \\ -(c - \boldsymbol{b}^\mathsf{T}\boldsymbol{A}^{-1}\boldsymbol{b})^{-1}\boldsymbol{b}^\mathsf{T}\boldsymbol{A}^{-1} & (c - \boldsymbol{b}^\mathsf{T}\boldsymbol{A}^{-1}\boldsymbol{b})^{-1} \end{pmatrix}.$$

**Lemma 13** (Sherman-Morrison).

$$(\boldsymbol{A} + \boldsymbol{b}\boldsymbol{b}^\mathsf{T})^{-1} = \boldsymbol{A}^{-1} - \frac{\boldsymbol{A}^{-1}\boldsymbol{b}\boldsymbol{b}^\mathsf{T}\boldsymbol{A}^{-1}}{1 + \boldsymbol{b}^\mathsf{T}\boldsymbol{A}^{-1}\boldsymbol{b}} \quad and \quad (\boldsymbol{A} - \boldsymbol{b}\boldsymbol{b}^\mathsf{T})^{-1} = \boldsymbol{A}^{-1} + \frac{\boldsymbol{A}^{-1}\boldsymbol{b}\boldsymbol{b}^\mathsf{T}\boldsymbol{A}^{-1}}{1 - \boldsymbol{b}^\mathsf{T}\boldsymbol{A}^{-1}\boldsymbol{b}}.$$

## B  Proofs

*Proof of Lemma 9.*  Recall that

$$h := \frac{2}{1 + 2\lambda + \sqrt{1 + 4\lambda}}.$$

This allows us to write

$$\begin{aligned}
h_t &= \frac{\cosh\big((t+1)\nu\big) - \cosh\big((1-t)\nu\big)}{2\lambda \sinh(\nu) \sinh\big((t+1)\nu\big)} \\
&= \frac{1 - \left(\frac{2\lambda}{1+2\lambda+\sqrt{4\lambda+1}}\right)^{2t}}{\sqrt{4\lambda+1} + \frac{2\lambda^2}{1+2\lambda+\sqrt{4\lambda+1}}\left(1 - \left(\frac{2\lambda}{1+2\lambda+\sqrt{4\lambda+1}}\right)^{2t}\right)} \\
&= \frac{1 - (\lambda h)^{2t}}{\sqrt{4\lambda+1} + \lambda^2 h(1 - (\lambda h)^{2t})}.
\end{aligned}$$

Verifying that $1/h = \sqrt{4\lambda+1} + \lambda^2 h$ and plugging this in yields

$$h_t = \frac{1 - (\lambda h)^{2t}}{1 - (\lambda h)^{2t+2}} h,$$

as desired. $\qquad\square$

**Claim 14.** *We claimed that*

$$\sum_{t=1}^{T} c_t \geq h \int_0^{T-1} \int_{t+1}^{T} (\lambda h)^{2(k-t)} \frac{1 - (\lambda h)^{2t}}{1 - (\lambda h)^{2(t+1)}} \, dk \, dt = O\left(\frac{-hT}{(\lambda h)^2 \log(\lambda h)}\right).$$

*Evaluating the integral gives*

$$\begin{aligned}
&\int_1^{T-1} \int_{t+1}^{T} (\lambda h)^{2(k-t)} \frac{1 - (\lambda h)^{2t}}{1 - (\lambda h)^{2(t+1)}} \, dk \, dt \\
&= -\frac{T}{\log((\lambda h)^2)} + \frac{(\lambda h)^2 - \log((\lambda h)^2)}{\log^2((\lambda h)^2)} + \frac{1}{\log((\lambda h))^2} \\
&\quad + \frac{(\lambda h)^{2T} + ((\lambda h)^2 - 1)(\lambda h)^{2T}((\lambda h)^{2T} - 1)\log((\lambda h)^{-2} - (\lambda h)^2)}{(\lambda h)^2 \log^2((\lambda h)^2)} \\
&\quad - \frac{((\lambda h)^2 - 1)(\lambda h)^{2T} - 1)\log((\lambda h)^{-2T+2} - (\lambda h)^2)}{\log^2((\lambda h)^2)} \\
&= -\frac{T}{2\log(\lambda h)} + o(1).
\end{aligned}$$

## B.1 Proof of Theorem 2

*Proof.* First, we consider $\alpha \leq 0$. The objective is concave in $x$ so the inner maximum is found at zero derivative, i.e. at $x = \frac{a-b}{\alpha}$ (it remains to check this is feasible) where the value equals $V^* = \min_{a \in \mathbb{R}^d} \|a\|^2 - \frac{\|a-b\|^2}{\alpha}$. This optimization is convex in $a$ (again owing to the sign of $\alpha$) and again minimized at vanishing derivative. That is, $a = \frac{b}{1-\alpha}$, (back substitution reveals $x = \frac{b}{1-\alpha}$, which is indeed feasible since $\|b\| \leq 1$ by assumption and we are in the case $\alpha \leq 0$). Thus, the value equals $V^* = \frac{\|b\|^2}{1-\alpha}$, as claimed.

Next, we consider $\alpha \geq 0$. The objective is convex in $x$ and must take its maximum on the boundary. Specifically, it is maximized at the point of the unit sphere maximizing the linear term, i.e. at $x = \frac{b-a}{\|b-a\|}$. Plugging this in, we find value equal to $V^* = \min_{a \in \mathbb{R}^d} \|a\|^2 + \alpha + 2\|b - a\|$. This is a concave function of $a$, although it is not differentiable due to the square-less norm. Yet it is minimized where zero is a sub-gradient (the unit ball is sub-gradient to the norm at $\mathbf{0}$). In our case this happens at $a = b$, where we used the assumption that $\|b\| \leq 1$. We conclude that the value is $V^* = \|b\|^2 + \alpha$ as desired. $\square$

## B.2 Proof of Theorem 7

*Proof.* By induction. For the base case notice that $\boldsymbol{R}_T$ is the $D$-banded matrix $\boldsymbol{I} + \lambda \boldsymbol{K}$. Now assume that $\boldsymbol{R}_t^{-1}$ is a sum of a $D$-banded matrix and an $N$-blocked matrix. Naturally $D, V \geq 1$ and $N \geq 0$. Then $\boldsymbol{b}_t'$ is $\max\{D-1, N-1\}$-sparse, and hence $\boldsymbol{b}_t' \boldsymbol{b}_t'^{\mathsf{T}}$ is $\max\{D-1, N-1\}$-blocked. Also $\boldsymbol{u}_t$ is $(V-1)$-sparse, and $\boldsymbol{A}_t'$ is $D$-banded plus $(N-1)$-blocked. So $\boldsymbol{A}_t' \boldsymbol{u}_t$ is $\max\{D+V-2, N-1\}$-sparse. Now $\boldsymbol{A}_t' \boldsymbol{u}_t \boldsymbol{b}_t'^{\mathsf{T}}$ is $\max\{N-1, D+V-2\}$-blocked. So $\boldsymbol{R}_{t-1}^{-1}$ is $D$-banded plus $\max\{N-1, D+V-2\}$-blocked. So if $N \geq D+V-2$ it remains $D$-banded plus $N$-blocked. (Note that strict inequality is wasteful: after a few pullbacks $\boldsymbol{R}_s^{-1}$ will be $D$-banded plus $(D+V-2)$-blocked). $\square$

## B.3 Proof of Theorem 11

*Proof.* By induction. The base case $t = T$ follows by Theorem 1 and noticing that $\gamma_T = 0$. Then

$$
\boldsymbol{R}_t = \begin{pmatrix} \boldsymbol{I} + \lambda \boldsymbol{K}_{t-1} & -\lambda \boldsymbol{e}_{t-1} \\ -\lambda \boldsymbol{e}_{t-1}^{\mathsf{T}} & 1 + 2\lambda + \gamma_t \end{pmatrix}^{-1} = \begin{pmatrix} (\boldsymbol{I} + \lambda \boldsymbol{K}_{t-1})^{-1} + \lambda^2 c_t \boldsymbol{z}_{t-1} \boldsymbol{z}_{t-1}^{\mathsf{T}} & \lambda c_t \boldsymbol{z}_{t-1} \\ \lambda c_t \boldsymbol{z}_{t-1}^{\mathsf{T}} & c_t \end{pmatrix}
$$

where used that $c_t = \frac{1}{\frac{1}{h_t} + \gamma_t} = \frac{1}{1 + 2\lambda + \gamma_t - \lambda^2 h_{t-1}}$. Now using that $c_t \geq 0$, Theorem 5 gives optimal prediction $\boldsymbol{a}_t = \lambda c_t \boldsymbol{X}_{t-1} \boldsymbol{z}_{t-1}$ and the update equation (7) gives $\boldsymbol{R}_{t-1} = (\boldsymbol{I} + \lambda \boldsymbol{K}_{t-1})^{-1} + \lambda^2 c_t \boldsymbol{z}_{t-1} \boldsymbol{z}_{t-1}^{\mathsf{T}} + \lambda^2 c_t^2 \boldsymbol{z}_{t-1} \boldsymbol{z}_{t-1}^{\mathsf{T}}$. By Sherman-Morrison (Lemma 13), using that $(\boldsymbol{I} + \lambda \boldsymbol{K}_t) \boldsymbol{z}_t = \boldsymbol{e}_t$,

$$
\boldsymbol{R}_{t-1}^{-1} = \boldsymbol{I} + \lambda \boldsymbol{K}_{t-1} - \frac{(\lambda^2 c_t + \lambda^2 c_t^2)(\boldsymbol{I} + \lambda \boldsymbol{K}_{t-1}) \boldsymbol{z}_{t-1} \boldsymbol{z}_{t-1}^{\mathsf{T}} (\boldsymbol{I} + \lambda \boldsymbol{K}_{t-1})}{1 + (\lambda^2 c_t + \lambda^2 c_t^2) \boldsymbol{z}_{t-1}^{\mathsf{T}} (\boldsymbol{I} + \lambda \boldsymbol{K}_{t-1}) \boldsymbol{z}_{t-1}}
$$

$$
= \boldsymbol{I} + \lambda \boldsymbol{K}_{t-1} - \frac{(\lambda^2 c_t + \lambda^2 c_t^2) \boldsymbol{e}_{t-1} \boldsymbol{e}_{t-1}^{\mathsf{T}}}{1 + (\lambda^2 c_t + \lambda^2 c_t^2) h_{t-1}}
$$

witnessing the expression for

$$
\gamma_{t-1} = -\frac{\lambda^2 c_t (1 + c_t)}{1 + \lambda^2 c_t (1 + c_t) h_{t-1}}.
$$

We finally check, using $h_t = \frac{1}{1 + 2\lambda - \lambda^2 h_{t-1}}$, that

$$
c_{t-1} = \frac{1}{\frac{1}{h_{t-1}} - \frac{\lambda^2 c_t (1+c_t)}{1 + \lambda^2 c_t (1+c_t) h_{t-1}}} = h_{t-1} + \lambda^2 h_{t-1}^2 c_t (1 + c_t). \qquad \square
$$

## C  Inverse Update Equation

The update equation (7) for $\boldsymbol{R}_t$ is simple, but results in dense matrices $\boldsymbol{R}_t$, even though $\boldsymbol{R}_t^{t-1}$ is sparse. In this section we derive a recursion for $\boldsymbol{R}_t^{-1}$ purely in terms of inverses. Let us decompose

$R_t^{-1} = \begin{pmatrix} A'_t & b'_t \\ b'^{\mathsf{T}}_t & c'_t \end{pmatrix}$. Then

$$A_t = A'^{-1}_t + A'^{-1}_t b'_t (c'_t - b'^{\mathsf{T}}_t A'^{-1}_t b'_t)^{-1} b'^{\mathsf{T}}_t A'^{-1}_t \tag{11a}$$

$$b_t = - A'^{-1}_t b'_t (c'_t - b'^{\mathsf{T}}_t A'^{-1}_t b'_t)^{-1} \tag{11b}$$

$$c_t = (c'_t - b'^{\mathsf{T}}_t A'^{-1}_t b'_t)^{-1} \tag{11c}$$

If $c_t \geq 0$ it follows from (7) that

$$R_{t-1} = A'^{-1}_t + [b_t \; u_t] \begin{pmatrix} 1 + \frac{1}{c_t} & -c_t \\ -c_t & c_t(c_t - 1) \end{pmatrix} [b_t \; u_t]^{\mathsf{T}}$$

and hence

$$
\begin{aligned}
R_{t-1}^{-1} &= A'_t - A'_t [b_t \; u_t] \left( \begin{pmatrix} 1 + \frac{1}{c_t} & -c_t \\ -c_t & c_t(c_t - 1) \end{pmatrix}^{-1} + [b_t \; u_t]^{\mathsf{T}} A'_t [b_t \; u_t] \right)^{-1} [b_t \; u_t]^{\mathsf{T}} A'_t \\
&= A'_t - [-c_t b'_t \; A'_t u_t] \left( \begin{pmatrix} c_t(1 - c_t) & -c_t \\ -c_t & -(1 + \frac{1}{c_t}) \end{pmatrix} + [b_t \; u_t]^{\mathsf{T}} A'_t [b_t \; u_t] \right)^{-1} [-c_t b'_t \; A'_t u_t]^{\mathsf{T}} \\
&= A'_t - \frac{[-c_t b'_t \; A'_t u_t] \begin{pmatrix} u_t^{\mathsf{T}} A'_t u_t - (1 + \frac{1}{c_t}) & c_t(b'^{\mathsf{T}}_t u_t + 1) \\ c_t(b'^{\mathsf{T}}_t u_t + 1) & c_t^2(c'_t - 1) \end{pmatrix} [-c_t b'_t \; A'_t u_t]^{\mathsf{T}}}{\left( u_t^{\mathsf{T}} A'_t u_t - (1 + \frac{1}{c_t}) \right) c_t^2(c'_t - 1) - c_t^2(b'^{\mathsf{T}}_t u_t + 1)^2} \\
&= A'_t - \frac{[b'_t \; A'_t u_t] \begin{pmatrix} u_t^{\mathsf{T}} A'_t u_t - (1 + \frac{1}{c_t}) & -b'^{\mathsf{T}}_t u_t - 1 \\ -b'^{\mathsf{T}}_t u_t - 1 & c'_t - 1 \end{pmatrix} [b'_t \; A'_t u_t]^{\mathsf{T}}}{\left( u_t^{\mathsf{T}} A'_t u_t - (1 + \frac{1}{c_t}) \right) (c'_t - 1) - (b'^{\mathsf{T}}_t u_t + 1)^2}
\end{aligned}
$$

Which expands to

$$
\begin{aligned}
R_{t-1}^{-1} = A'_t &- \frac{\left( u_t^{\mathsf{T}} A'_t u_t - (1 + \frac{1}{c_t}) \right) b'_t b'^{\mathsf{T}}_t}{Z} \\
&+ \frac{(b'^{\mathsf{T}}_t u_t + 1)(b'_t u_t^{\mathsf{T}} A'_t + A'_t u_t b'^{\mathsf{T}}_t)}{Z} - \frac{(c'_t - 1) A'_t u_t u_t^{\mathsf{T}} A'_t}{Z}
\end{aligned}
$$

where we abbreviated $Z = \left( u_t^{\mathsf{T}} A'_t u_t - (1 + \frac{1}{c_t}) \right) (c'_t - 1) - (b'^{\mathsf{T}}_t u_t + 1)^2$. On the other hand for $c_t \leq 0$ we find

$$R_{t-1}^{-1} = \left( A'^{-1}_t + \frac{b_t b_t^{\mathsf{T}}}{c_t} + \frac{b_t b_t^{\mathsf{T}}}{1 - c_t} \right)^{-1} = A'_t + \frac{b'_t b'^{\mathsf{T}}_t}{1 - c'_t}.$$

## C.1 The $c_t$

In Figure 2 we graph the $c_t$, which are crucial in determining the regret, for $4 \times 3$ natural choices the regularization matrix $K$ and the data constraints $v_t$. We consider minor variations of the standard second-order regularization (2), where we choose to regularize either end-point down to zero. We consider first order $v_t = e_t$, second order $v = e_t - e_{t-1}$ and third order $v_t = e_t - 2e_{t-1} + e_{t-2}$ data constraints. We see that the first two never hit negative $c_t$, whereas the third option consistently runs into them.

(a) Norm of data $v_t = e_t$

(b) Norm of increments $v = e_t - e_{t-1}$

(c) Norm of increments of increments $v_t = e_t - 2e_{t-1} + e_{t-2}$

Figure 2: Graphs of the $c_t$ as a function of $t$ for $T = \lambda = 128$ under the assumption that the data are norm bounded (a), increment bounded (b) and increments of increments bounded (c). Note the logarithmic scale on the first two plots and the linear scale on the last plot, where the $c_t$ go negative. Each plot shows four lines, one for each possibility of regularizing the comparator down to zero at either end-point.