[Reviews · NeurIPS 2015]

Submitted by Assigned_Reviewer_1

This paper provides the minimax solution to the online vector prediction problem. The proposed algorithm is efficient with amortized O(d) time per round, where d is the dimension of the data. The regret grows as

T/\sqrt{\lambda}, where \lambda is the smoothness of the comparator sequence.

The paper is well-written and analyses are clear. The proposed algorithm and regret bounds are novel. Below are some of my questions and concerns:

1) The regret has a linear dependence on T, unlike many other online learning problems that usually have a sublinear dependence. This may suggest that the comparator sequence is too strong. It would be interesting to see how adding additional assumptions affects the regret bound.

2) In equation (1), why there is no complexity term for the learner (i.e. \sum_t ||a_t - a_t-1||^2)?

3) When \lambda approaches infinity, the comparator sequence would become constant, making (1) to be the standard online learning problem. But then the regret approaches 0 (T/\sqrt{\lambda}). This seems unintuitive.

4) In (1) The third term contains \hat{a}^{T+1}, which is undefined.

Summary: This paper provides the minimax solution to the online vector prediction problem and showed an regret bound. The regret bound is tight, but it has a linear dependence on T, the length of the game, which seem not very helpful in practice.

Submitted by Assigned_Reviewer_2

The biggest weakness of the paper is the experiment results section: 1. The results only appear in the supplemental material. There are no results in the main paper. 2. Although enlightening, the results are only on univariate and synthetic data. The univariate part is disappointing since the authors make a lot of effort to make sure the method works in a multivariate setting. 3. There is not much comparison to other methods, despite the huge library of time series prediction methods out there. 4. There are only graphical results, no quantitative performance results.

Other weaknesses are: 1. By studying the equations, there is an implicit assumption that all dimensions in the data are of the same units. This will often not be the case in many real multivariate time series problems. At the very least, this should be made explicit. 2. The setup seems a bit contrived, there is an assumption that we can place a worst case bound (in an adversarial setting) on the Euclidean norm of the data. When would a problem naturally have such a bound? I can imagine real problems where we might have an infinity norm bound on the data (because we know the upper and lower limits of each time series), but what about L2?

A more open ended question is that both the L2 norm constraint on the data and the choice of (L2) smoothness penalty on the comparator seem to be a bit arbitrary. Do these L2 constraints have seem similar effects to adding a Gaussian prior in the appropriate locations? Maybe the authors could elaborate in the paper on how one would choose what penalty to place on the comparator.

Other areas the authors could elaborate on are: 1. Why is the smoothness penalty only applied to the comparator? What would happen if both the learner and the comparator had a smoothness penalty? Would the results be desirable? 2. How would the results differ if the outer min and max in (1) were not alternating, but all min then all max? How would the results be different?

The mathematical derivations are dense but appear to be correct. Maybe modifying some of the notation in places like (7) and making explicit the dependence of L* on X would help. Since the authors already have a supplemental material section, maybe they could clarify some of the derivations. Like in Thm 1 they utilize the relation between trace(A'*B) and sum(sum(A.*B)), but don't mention that due to the space constraints. Maybe some elaboration in supplemental material would be nice.
Summary: This paper proposes a minimax framework for predicting time series using square loss. It mimics a lot of the work that has been done in minimax time series prediction in MDL, but extends it to consider the square loss as opposed to log loss.

Submitted by Assigned_Reviewer_3

The paper solves an online time series prediction problem with the L2 norm featured heavily, to measure square loss, regularization, and data constraints. The usual backward induction method is employed to solve this repeated game, with theoretically interesting results. The proofs are straightforward though tedious. The paper is very clear; the proofs follow a linear ordering for the most part, with each result building upon the last towards the minimax solution of the game presented initially.

The problem is a very well-studied one with widespread applications in econometrics, finance, and other fields. These applications certainly suggest an adversarial formulation, but previous practical approaches to the problem generally just posit a model without justification, other than that it works (e.g. ARMA, GARCH; see also work of Anava et al. which isn't cited). The analysis raises more (interesting) questions than it answers, particularly because of the lack of obvious interpretations of it. It can be seen as a natural follow-up to [9, 13], in the context of time series prediction it is new but leads to phenomena with much prior applied precedent, like autoregression with shrinkage.

The analysis itself is nicely executed but not particularly original. But the formulation is original, and is suited enough to the problem that the paper definitely "deserves to be published" because there is not much work of this type on this very worthwhile problem. The robustness of the solution to ball radius is interesting in this context (typo on line 161: "the figure to the right"), though it is clear enough from the one-shot game that it should have the same d-dependence as the squared norm of x.

A few specific comments: - It would be nice to see experiments on real data, and it is puzzling to see them absent besides a few simulations. Woodbury formulas are liberally applied in the appendix to demonstrate a tractable update, and I am very curious how well this strategy performs empirically in the unconstrained case. This is particularly true for real time series where the data might reasonably be from a slowly drifting distribution, and therefore would be "nearly" in the unit ball after rescaling. - I'd like to see a bit more discussion of the ARMA/GARCH models (a standard reference outside learning is the book by the econometrist Hamilton). This minimax solution ends up being autoregressive, with coefficients determined by the data somewhat like in [9], while the coefficients for e.g. ARMA are often fit with a linear regressor, so I am curious if you think there is a link here, though it is unclear what ARMA models mean with an adversarial sequence. Any link would be valuable because the linear regression could be sharpened with constraints, etc. more transparently than this repeated game. - The choice of L2 norm is a bit dissatisfying. On a high level it seems to be because L2 is self-conjugate, and therefore the induction steps can be chained together this way; I have no suggestion for how to do it otherwise. L1/L_\infty would be nice to see somehow though, because the (nearly) Toeplitz K matrices being used here are also in vogue for trend filtering in a very similar problem context.
Summary: The authors give the exact minimax solution of an online time series prediction problem with square loss and also square L2-norm regularization and constraints. The solution is novel and surprisingly efficient to compute, and adds significant insight by exhibiting properties of popularly used models for these tasks; however, experiments are essentially not included for this practical problem.

Submitted by Assigned_Reviewer_4

The paper is very well-written, both syntactically and logically: the ideas were easy to follow and the contribution is quite clear. The idea to analyze arbitrarily generated time series using the regret criteria is interesting in my opinion, yet it is not the first time such analysis is made (despite the fact that the authors do not cover the relevant literature in their related work section). I expect the authors to do a better literature review if the paper is accepted.

One issue that bothers me the most is the non-standard definition of the regret: why should the comparator class be penalized on its smoothness complexity? In particular, many time series are not smooth and yet are simply predicted (for example, AR type time series). In my opinion, the logical thing to do is to restrict the model and not the predictions. For example, if we consider an AR model, the restriction should be on the smoothness of the AR coefficients and not on the smoothness of the predictions. The current regularization might hold as well, but needs to be better motivated by the authors.

The authors mention that their setting can handle with ARMA models, which seems misleading to me. I am not sure what the authors mean here - if the time series is allowed to be arbitrarily generated (as the authors assume) then ARMA models are not well-defined (see the work of Anava et al (2013)). If the time series is stochastically generated (and is in fact an ARMA process), then a regret result is meaningful, but I expect the authors to clarify this point. In particular, such results should be better contrasted with existing results (both theoretically and empirically).

I find the experimental section very poor (even though it appears only in the appendix). In particular, the authors do not compare their result to any baseline, error plots are not presented, the setting and the generation mechanism of the time series are not specified, and more. I suggest the authors to remove it completely or follow the standard requirements in this field.

Despite the weaknesses I spot, I vote for acceptance and willing to upgrade my score according to the authors response (mainly motivating the regret definition).

minor comments: (1) line 282 - sill -> still. (2) I would use similar notations in section 3 and 4 - \alpha should be c, \beta should be w, and c should be u. It will be easier to follow this way. (3) please use the word "intriguing" less than you do.

UPDATE: The authors response did not allay my concerns regarding the regret formulation, and thus I chose not to upgrade my score. However, since the authors suggest a new approach for time series prediction, I think the paper should be accepted despite its weak theory and experiments. In fact, I am quite sure that this approach will be far behind existing approaches in practice, but might open the door for interesting future work.

Missing literature: I would start with the standard textbooks (Hamilton, Box&Jenkins, Brockwell&Davis), and advance to newer papers that combine learning techniques for time series prediction (some even use the regret criterion). Some pointers: Generalization Bounds for Time Series Prediction with Non-stationary Processes, Nonparametric Time Series Prediction Through Adaptive Model Selection, Online Prediction of Time Series Data With Kernels, Online Learning for Time Series Prediction.

Summary: The paper addresses the problem of time series prediction in an adversarial setting, that is, the time series is not assumed to comply with any statistical model. The main result is a minimax regret in this setting, but the comparator class is penalized on the smoothness complexity of its chosen predictions. The main technical contribution of the paper is an efficient method to compute the minimax result, which follows by the structure of the problem: the quadratic minimax problem at each step is either concave or convex, and analytically solvable in both cases. The authors present experiments with synthetic data.

Author Feedback
Author rebuttal: We would like to thank all the reviewers for their feedback. First,
we'll address some common points:

First, we certainly need to revise the motivation for the inclusion of
the L2 comparator smoothness complexity in the regret (1). This notion
of regret is standard in online learning, going back at least to
Tracking the Best Linear Predictor, JMLR 2001 by Herbster and Warmuth,
who view it as the natural generalization of L2 regularization to deal
with non-stationarity comparators. We will add this fact to the paper.
In the paper we provided additional motivation for (1) by interpreting
the complexity term as the magnitude of the noise required to generate
the comparator using a multivariate Gaussian random walk, and,
generalizing slightly, as the energy of the innovations required to
model the comparator using a single, fixed linear time series model
(e.g. specific ARMA coefficients). However, we now see that this may
have incorrectly suggested that we were trying to compete with a class
of time series models. We will revise this additional motivation to
eliminate any such potential confusion and refer to work that
addresses this worthwhile task. Finally, for completeness, we will
mention the related approach of putting a hard constraint on the
comparator complexity and compare it to our (1), which is akin to the
constrained optimization Lagrangian.

Second, a common misconception was that our regret expression of the
form T/\sqrt{\lambda} corresponds to linear regret. It is natural to
allow the coefficient \lambda to grow with T. A reasonable setting,
suggested by the analysis, would be \lambda=T, resulting in \sqrt(T)
regret.

The experiments can rightly be criticized. They add little to the main
point: that the algorithm is optimal for the prediction game that we
consider. We will follow Reviewer 9's advice and remove this section.
Further work will include a closer look at quantifying the improvement
over other algorithms.

Reviewer 3: We feel that there is some confusion about the results of
this paper. The main contribution was deriving the exact minimax
algorithm for a new class of games on sequences. Online learning with
adversarial data is usually tackled by: proposing an algorithm,
proving an upper bound on the regret, and showing a matching lower bound
of the same order. In contrast, finding the minimax algorithm is a
much harder task as we must solve the nested optimization problem (1)
by specifying the optimal strategy for both players at every
round. The resulting algorithm has the strongest guarantee: we can
cause any other algorithm at least as much regret. It's still
uncommon to find minimax algorithms, and we think their study is
important.

In light of the strictness of minimax optimality, we can explain the choice
of L2 regularization: other norm regularizations do not lead to tractable
minimax algorithms; the value function increases in complexity under the
backwards induction.

"other areas:" 1. Good question. We tried this setting and the learner
becomes even more conservative. No qualitative change results, and
hence we opted for the simplest exposition. We will add a comment.
2. Without the alternating min/max, the learning problem degenerates into a
game with a single round.

Reviewer 4:
Competing with ARMA models and using other norms besides L2 are both
interesting topics for further work that go beyond this paper. We will
mention them in the discussion. Thanks for your comments.

Reviewer 6:
Please see the comments above concerning the definition of regret and the
minimax regret rates.

Reviewer 8:
2) See response to Reviewer 3 above.

3) This is not quite correct: the first and last smoothness terms
still penalize the constant comparators. Hence as \lambda tends to
\infty the only affordable comparator is identically zero. The minimax
regret compared to that single sequence is indeed zero.
If we would not charge for squared norm of first and last comparator
elements then we would indeed recover the standard online learning
problem, for which the regret is of order \ln T. The derivation of the
regret bound in Section 5 would change quite subtly. We presented the
version with the simpler matrix and recurrence.

4) Thanks, we'll add \hat{a}_0 = \hat{a}_{T+1} = 0.

Reviewer 9: We will be more thorough in our literature review;
pointers are welcome. Please see the comments above regarding the
definition of regret, the use of an ARMA model in defining the
complexity penalty, and the experimental section. Thanks for the other
suggestions; we will incorporate them.